spectroscopy/analytical chemistry/photochemistry

serum albumins, pentamethine cyanine dyes, globular proteins, far-red fluorescent probes, molecular docking

**Author for correspondence:**
V. Kovalska
e-mail: v.kovalska@gmail.com

This article has been edited by the Royal Society of Chemistry, including the commissioning, peer review process and editorial aspects up to the point of acceptance.

# Far-red pentamethine cyanine dyes as fluorescent probes for the detection of serum albumins

D. Aristova[1], G. Volynets[1], S. Chernii[1], M. Losytskyy[1], A. Balanda[1], Yu. Slominskii[2], A. Mokhir[3], S. Yarmoluk[1] and V. Kovalska[1,4]

[1]Institute of Molecular Biology and Genetics NASU, 150 Zabolotnogo Street, 03143 Kyiv, Ukraine
[2]Institute of Organic Chemistry NASU, 5 Murmans'ka Street, 02094 Kyiv, Ukraine
[3]Organic Chemistry II, Friedrich-Alexander-University of Erlangen-Nuremberg, Nikolaus-Fiebiger-Strasse 10, 91058 Erlangen, Germany
[4]Scientific Services Company Otava Ltd, 150 Zabolotnogo Street, 03143 Kyiv, Ukraine

VK, 0000-0001-8305-9398

Benzothiazole based cyanine dyes with bridged groups in the pentamethine chain were studied as potential far-red fluorescent probes for protein detection. Spectral-luminescent properties were characterized for unbound dyes and in the presence of serum albumins (bovine (BSA), human (HSA), equine (ESA)), and globular proteins (β-lactoglobulin, ovalbumin). We have observed that the addition of albumins leads to a significant increase in dyes fluorescence intensity. However, the fluorescent response of dyes in the presence of other globular proteins was notably lower. The value of fluorescence quantum yield for dye bearing a sulfonate group complexed with HSA amounted to 42% compared with 0.2% for the free dye. The detection limit of HSA by this dye was greater than 0.004 mg ml$^{-1}$ which indicates the high sensitivity of dye to low HSA concentrations. Modelling of structure of the dyes complexes with albumin molecules was performed by molecular docking. According to these data, dyes could bind to up to five sites on the HSA molecule; the most preferable are the haemin-binding site in subdomain IB and the dye-binding site in the pocket between subdomains IA, IIA and IIIA. This work confirms that pentamethine cyanine dyes could be proposed as powerful far-red fluorescent probes applicable for highly sensitive detection of albumins.

# 1. Introduction

Albumins are multifunctional proteins present in the blood serum of animals. Owing to their conformational flexibility, they are major transport proteins in blood plasma for many endogenous and exogenous compounds like drugs, hormones, fatty acids and other bioactive small molecules. Several mammalian and avian serum albumins (SAs) are allergens [1]. Also, the concentration of SA in the blood is a diagnostic parameter that can indicate many diseases [2].

Currently, there are a number of standard colorimetric methods for the detection of albumins, like the Lowry method and the biuret method. However, they have several disadvantages, among which are low sensitivity, narrow linear range, difficult analysis procedures, etc. [3]. Therefore, interest increases in fluorescent detection methods because of their convenience, sensitivity, simplicity of the technique, etc. There are many fluorescent dyes with different structures, which can be successfully used for the detection and visualization of various biomolecules. Recent studies include, for example, a fluorescent probe with a fluorescein-based fluorophore for detection of thionitrous acid in aqueous medium and cells [4], and a sulfilimine-based fluorescent probe for the specific detection and quantifying of hypobromous acid [5]. It is also worth mentioning the design of a two-photon styryl fluorescent probe for real-time monitoring the changes in the activity of acetylcholinesterase in the brain *in vivo* [6]. Also, near-infrared (NIR) fluorescent turn-on probes were developed for the detection and visualization intracellular $H_2S$ [7]. A number of fluorescent dyes for protein determination in solution and monitoring of protein structure modification are already commercially available [8]. One of these dyes is a well-known hydrophobic fluorescence probe for protein detection 1-anilinonapthalene-8-sulfonic acid (1,8-ANS) [9]. The bovine serum albumin (BSA) globule is known to be able to bind (at pH values between 5 and 10) five molecules of 1,8-ANS resulting in ANS fluorescent response [10,11]. Another well-known dye for protein characterization is Nile Red, which can be applied for monitoring conformational changes of different proteins. The dye 9-(dicyanovinyl)-julolidine has been used as a probe to detect the formation of hydrophobic microdomains, e.g. in protein aggregation [9]. Also, there are a number of dyes for quantification of protein in solution, such as eosin B and eosin Y [12], and Albumin Blue [13]. In the previous study, we have shown that squaraine dyes are efficient noncovalent labels for albumin detection, exhibiting high fluorescence quantum yields when bound to these proteins [14–16]. 2-Quinolone and coumarin polymethines were shown to be able to increase the fluorescence intensity one hundred times with bright emission in the presence of albumin [17]. Cyanine dyes are studied as fluorescent probes for proteins and nucleic acids detection and visualization owing to their favourable spectral characteristics, namely, the possibility to vary the absorption and emission wavelengths (reaching far-red and even the NIR range), large extinction coefficients, high fluorescence quantum yields, etc. [18].

The probes, which can be excited and which can emit in far-red and NIR regions, are especially useful for direct detection of albumins in biological samples, because most of the tissues are transparent to this light and do not exhibit any autofluorescence in this spectral region. [8]. NIR and far-red dyes, such as heptamethine cyanines and squaraines, were previously reported as protein-sensitive light-up probes [19,20]. Far-red pentamethine cyanine dyes are used as labels and probes for the staining of biological molecules [21]. Pentametnine cyanines with the cycle (bridge) in the polymethine chain are used in various fields, for example, as ingredients of photographic elements [22]. As an advantage of bridged pentametnine cyanines over 'longer-chain' heptamethine cyanines, the higher quantum yield of the first ones [21,23] could be considered. Previously, we have shown that far-red bridged pentamethine dyes with cyclohexene groups in the polymethine chain increase their fluorescence intensity dozens of times in the presence of BSA [24]. As a continuation of these studies, in the current work, we examined a series of bridged pentamethine cyanine dyes based on various heterocycles (figure 1) as probes for the detection of SAs. In particular, the spectral-luminescent properties of four pentamethine cyanines were characterized in the presence of BSA, human serum albumin (HSA) and equine serum albumin (ESA). We also investigated whether these dyes will also give the fluorescent response to other globular proteins of distinct structure including ovalbumin (OVA) and β-lactoglobulin (BLG). Zwitterionic **dye 3** was found to be the most sensitive to albumins. For this dye, the HSA detection range and affinity of binding were determined. To get a better understanding about the interaction of the dyes with proteins, we compared spectral-luminescent properties of these dyes to those of widely used hydrophobic dye 1,8-ANS. Finally, we have estimated the preferable binding sites of studied dyes on the HSA globule by using molecular docking.

**Figure 1.** Chemical structures of cyanine pentamethine dyes and 1,8-ANS.

**Scheme 1.** Outline of synthesis of the studied cyanine dyes.

# 2. Material and methods

## 2.1. Materials

Dimethyl sulfoxide (DMSO), methanol, acetonitrile, distilled water, 0.05 M Tris–HCl buffer (pH 7.9), 0.05 M phosphate buffer (pH 6.0) and 0.05 M Tris–HCl (pH 9.0) were used as solvents. HSA, BSA, ESA, OVA and BLG were purchased from Sigma-Aldrich (USA).

### 2.1.1. Synthesis of the dyes

The general scheme of the dyes synthesis is presented in scheme 1. Thiadicarbocyanine **dyes (1–4)** were obtained by condensation of quaternary salts of the corresponding 2-methylbenzothiazole derivatives with 1,3-diethoxy-5,5-dimethylcyclohexa-1,3-diene, as described in [25]. The structures of the dyes were confirmed using H1 NMR spectroscopy and LC–MS analysis. H1 nuclear magnetic resonance (NMR) spectra were recorded on a Varian Mercury VRX-400 spectrometer using DMSO-d6 as solvent and tetramethylsilane as internal standard. Liquid chromatography–mass spectra (LC–MS) analyses were performed using the Agilent 1100 LC/MSD SL (Agilent Technologies). Results of the dyes' characterization are provided below.

*Dye 1* (3-Ethyl-2-[3-(3-ethyl-5-methoxy-3H-benzothiazol-2-ylidenemethyl)-5,5-dimethyl-cyclohex-2-enylidene methyl]-5-methoxy-benzothiazol-3-ium iodide):

yield 36%; $^1$H NMR (400 MHz, DMSO-d$_6$): $\delta$: 1.06(6H, s), 1.31 (6H, t, $J = 7.0$ Hz), 2.61(4H, s) 3.88(6H, s) 4.46(4H, q, $J = 7.0$ Hz), 6.38(2H, b s), 7.05(2H, d, $J = 8.7$ Hz), 7.30(2H, s), 8.00(2H, d, $J = 8.5$ Hz). LC–MS: $m/z$ 519 [M]$^+$.

*Dye 2* (4-Methyl-cyclohexanesulfonate7-ethyl-6-[3-(7-ethyl-7H-[1,3]dioxolo[4′,5′:4,5]benzo[1,2-d]thiazol-6-ylidenemethyl)-5,5-dimethyl-cyclohex-2-enylidenemethyl]-[1,3]dioxolo[4′,5′:4,5]benzo[1,2-d]thiazol-7-ium 4-methylbenzenesulfonate):

yield 33%; [1]H NMR (400 MHz, DMSO-d$_6$): $\delta$: 1.04(6H, s), 1.28 (6H, t, $J = 7.0$ Hz), 2.27(3H, s), 2.56(4H, s) 4.38(4H, q, $J = 6.9$ Hz), 6.18(4 h, s), 6.29(2H, b s), 7.08(2H, d, $J = 7.9$ Hz), 7.45(2H, d, $J = 8.0$ Hz), 7.54(2H, s), 7.70(2H, b s)). LC–MS: $m/z$ 547 [M]$^+$.

**Dye 3** (2-{5,5-Dimethyl-3-[3-(3-sulfonate-propyl)-3H-benzothiazol-2-ylidenemethyl]-cyclohex-2-enylidenemethyl}-1-ethyl-naphtho[1,2-d]thiazol-1-ium):

yield 29%; [1]H NMR (400 MHz, DMSO-d$_6$): $\delta$: 1.09(6H, s), 1.70(2H, t, $J = 6.8$ Hz), 2.04(2H, m), 2.59(4H, s), 4.55(4H, t, $J = 6.8$ Hz), 4.87(4H, q, $J = 6.9$ Hz), 6.55(2H, b s), 7.37(1H, t, $J = 7.7$ Hz), 7.56(1H, t, $J = 7.9$ Hz), 7.75(3H, m), 8.01(1H, d, $J = 9.0$ Hz), 8.08(1H, m), 8.19(2H, m), 8.56(1H, d, $J = 9.1$ Hz) LC–MS: $m/z$ 603 [M + H]$^+$.

**Dye 4** (2-[5,5-Dimethyl-3-(1,3,4,9a-tetrahydro-2H-9-thia-4a-aza-fluoren-1-yl)-cyclohex-2-enylidenemethyl]-3-ethyl-benzothiazol-3-ium iodide):

yield 23%; [1]H NMR (400 MHz, DMSO-d$_6$): $\delta$: 1.11(6H, s), 1.32(3H, t, $J = 7.1$ Hz), 2.19(2H, m), 2.59(4H, m), 2.81(2H, t, $J = 6.0$ Hz), 4.27(2H, t, $J = 5.9$ Hz), 4.42(2H, q, $J = 7.2$ Hz), 6.47(1H, b s), 6.70(1H, b s), 7.37(2H, m), 7.58(2H, m), 7.70(2H, m), 8.04(2H, t, $J = 8.3$ Hz). LC–MS: $m/z$ 471 [M]$^+$.

## 2.2. Preparation of stock solutions of dyes and proteins

Pentamethine cyanines stock solutions were prepared by dissolving the weighted amount of the dyes at 2 mM concentration in DMSO. Stock solutions of proteins (HSA, BSA, ESA, OVA, BLG) were prepared by dissolving their weighted amounts in 0.05 M Tris–HCl buffer (pH 7.9) in a concentration equal to 0.2 mg ml$^{-1}$. Protein concentrations in stock solutions were equal to 3 µM for BSA, HSA (except for absorption spectra measurements where it was 6 µM), ESA, 4.5 µM for OVA and 11 µM for BLG.

## 2.3. Preparation of working solutions of dyes and proteins

Working solutions of free dyes were prepared by dilution of the dye stock solution in 0.05 M Tris–HCl buffer (pH 7.9), 0.05 M phosphate buffer (pH 6.0), 0.05 M Tris–HCl (pH 9.0), methanol, acetonitrile and DMSO. Working solutions of the dyes in the presence of proteins were prepared by the addition of the aliquot of the dye stock solution to the protein stock solution. The concentrations of dye working solutions amounted to 5 µM except for absorption spectra measurements where it was equal to 10 µM (in buffers and in the presence of HSA) and 1 µM (in methanol, acetonitrile and DMSO). All working solutions were prepared immediately before the experiments.

## 2.4. Spectral measurements

Spectroscopic measurements were performed in a standard quartz cuvette (10 × 10 mm). Fluorescence excitation and emission spectra were registered using the fluorescent spectrophotometer Cary Eclipse (Varian, Australia). Absorption spectra were registered using the spectrophotometers Genesys 20 (ThermoScientific) and Shimadzu UV-3600. All the spectral-luminescent characteristics of dyes were studied at room temperature.

## 2.5. Quantum yield determination

The quantum yield value of the **dyes 2** and **3** free and in the presence of SAs (HSA and BSA, respectively) was determined using Nile Blue (NB) solution in methanol as the reference (quantum yield value $\varphi_{NB} = 0.27$) [26]. Namely, solutions of **dye 2** and **dye 3** free, **dye 2** in the presence of BSA, **dye 3** in the presence of HSA (all in 0.05 M Tris–HCl buffer, pH 7.9) and NB in methanol were taken in such concentrations that their optical density values were equal at 645 nm. Fluorescence of all these solutions was excited at this wavelength (645 nm), and the area below each spectrum ($S_{dye}$ for **dyes 2** and **3** and $S_{NB}$ for NB) was calculated. Further, the fluorescence quantum yield $\varphi_{dye}$ of **dye 2** and **dye 3** free, **dye 2** in the presence of BSA, and **dye 3** in the presence of HSA was calculated as $\varphi_{dye} = \varphi_{NB} \times (S_{dye}/S_{NB}) \times (n_{H_2O}/n_{meth})^2$, where $n_{H_2O}$ and $n_{meth}$ are refractive indexes of water and methanol, respectively.

## 2.6. Molecular docking

Molecular docking of fluorescent dyes to the entire protein surface of HSA (crystal structure with PDB accession code 1AO6 [27]) and ESA (PDB ID: 4F5U [28]) was performed using CB-Dock web-server

(http://cao.labshare.cn/cb-dock/) [29]. Water molecules were removed from the PDB file of the receptor. Blind molecular docking was performed into five binding sites. The complexes were visually inspected using Discovery Studio Visualizer 4.0 [30].

## 2.7. Estimation of the affinity of **dye 3** binding to human serum albumin

The equilibrium constant of **dye 3** binding with HSA (binding constant, $K$) was estimated based on experimental measurements of the dependence of **dye 3** fluorescence intensity on HSA concentration in 0.05 M Tris–HCl buffer (pH 7.9). The experiment was performed three times, the average value of fluorescence intensity was calculated for each HSA concentration, the standard deviation was estimated. For calculation of $K$, we have started with the assumption that each protein globule has $n$ sites for dye binding. Though it is possible that binding with different sites is characterized with different values of $K$, for the rough estimation we considered the equilibrium constant of binding with different binding sites to have equal values. In this case, the following expression could be written:

$$K = \frac{C_{BL}}{(C_L - C_{BL}) \times (n \times C_P - C_{BL})}, \tag{2.1}$$

where $n$ is the number of dye molecules bound per HSA globule; $C_{BL}$, $C_P$ and $C_L$ are the concentrations of HSA-bound ligand (i.e. dye) molecules and total concentrations of protein globules and dye molecules, respectively. Solving the quadratic equation, we obtain:

$$C_{BL} = \frac{n \times C_P}{2} + \frac{C_L}{2} + \frac{1}{2 \times K} - \sqrt{\left(\frac{n \times C_P}{2} + \frac{C_L}{2} + \frac{1}{2 \times K}\right)^2 = n \times C_P \times C_L}. \tag{2.2}$$

To obtain the relation between $C_{BL}$ and the observed fluorescence intensity of the dye ($I$), the following considerations were taken into account. If a protein globule possesses several (more than one) site for the dye binding, the dyes bound to different sites could give different characteristics of fluorescence spectrum (i.e. maximum wavelength and quantum yield); this could be owing to different conformation and degree of internal motion restriction of the dyes bound in different sites. Meanwhile, our measurements showed that the shape and position of the maximum of the **dye 3** fluorescence spectrum does not depend on HSA concentration; thus, either the protein-bound dye molecules have similar conformation, or the molecules with other conformations make negligible contribution into the fluorescence spectrum. It is still possible that dye molecules bound to different sites have the same conformation but different quantum yield owing to different fixation strength. However (having in mind the mentioned complications), for the rough estimation of the binding constant we consider the dye fluorescence intensity to be proportional to the number of the bound dye molecules. In this case, the observed fluorescence intensity of the dye could be expressed as

$$I = \frac{C_L - C_{BL}}{C_L} \times I_0 + \frac{C_{BL}}{C_L} \times I_{max}, \tag{2.3}$$

where $I_0$ and $I_{max}$ are fluorescence intensities of the free dye in the absence of protein, and all dye molecules bound to the protein, respectively. It could be obtained from equation (2.3):

$$\frac{C_{BL}}{C_L} = \frac{I - I_0}{I_{max} - I_0}. \tag{2.4}$$

Together, equations (2.2) and (2.4) result in the final equation:

$$Y = A \times \left[\frac{1}{2} + \frac{x \times n}{2 \times C_L} + \frac{1}{2 \times K \times C_L} - \sqrt{\left(\frac{1}{2} + \frac{x \times n}{2 \times C_L} + \frac{1}{2 \times K \times C_L}\right)^2 - \frac{x \times n}{C_L}}\right], \tag{2.5}$$

where $x = C_P$, $Y = (I - I_0)$, $A = (I_{max} - I_0)$. Thus, the experimentally obtained curve of the dye fluorescence intensity is presented as the dependence of $(I - I_0)$ on $C_P$, and further fitted with the dependence (2.5). As a result of this fitting, the values of $K$, $n$ and $A$ are obtained as fitting parameters.

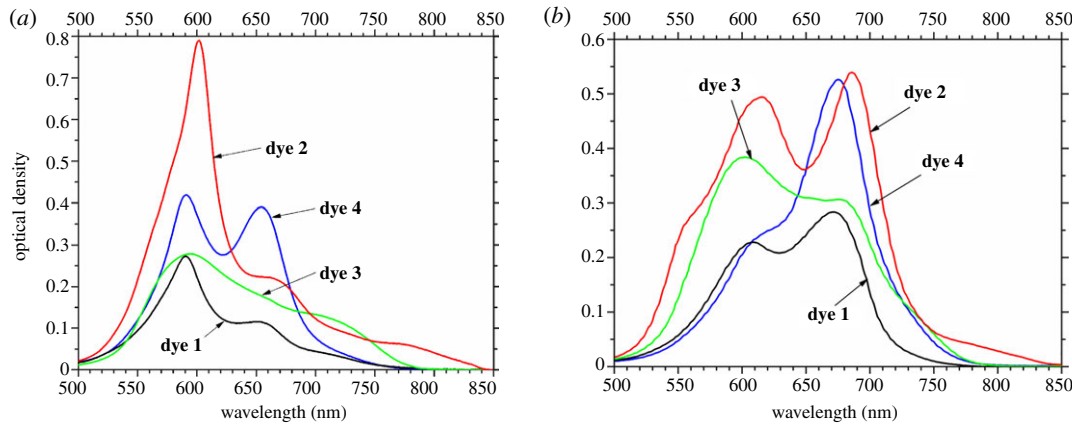

**Figure 2.** The absorption spectra of the studied cyanine dyes in 50 mM Tris–HCl (pH 7.9) buffer free (*a*) and in the presence of HSA (*b*). Dye concentration 10 µM, HSA concentration 0.4 mg ml$^{-1}$ (6 µM).

**Table 1.** Characteristics of the absorption spectra of the dyes in methanol, acetonitrile, DMSO and 50 mM Tris–HCl buffer, pH 7,9. ($\lambda_{max}$, maximum wavelength of absorption spectrum; $\varepsilon$, molar extinction coefficient at $\lambda_{max}$.)

| dye | methanol | | acetonitrile | | DMSO | | Tris–HCl buffer, pH 7.9 |
| | $\lambda_{max}$, nm | $\varepsilon$, $10^5$ M$^{-1}$ cm$^{-1}$ | $\lambda_{max}$, nm | $\varepsilon$, $10^5$ M$^{-1}$ cm$^{-1}$ | $\lambda$, nm | $\varepsilon$, $10^5$ M$^{-1}$ cm$^{-1}$ | $\lambda_{max}$, nm |
|---|---|---|---|---|---|---|---|
| **dye 1** | 662 | 0.78 | 660 | 0.63 | 673 | 0.62 | 592 |
| **dye 2** | 677 | 1.59 | 672 | 1.27 | 689 | 1.47 | 601 |
| **dye 3** | 666 | 1.32 | 663 | 1.15 | 676 | 1.06 | 594 |
| **dye 4** | 660 | 1.03 | 659 | 1.04 | 672 | 1.14 | 590/655 |

# 3. Results and discussion

## 3.1. Absorption spectra of the pentamethine cyanine dyes

Ultraviolet-visible absorption spectra of the studied dyes were acquired in organic solvents methanol, acetonitrile and DMSO (table 1, electronic supplementary material, figure S1). The absorption maxima of the cyanines in methanol and acetonitrile are rather close and located in the range of 660–677 nm and 659–663 nm, respectively, while the maxima in DMSO are shifted by about 10 nm to the long-wavelength region and lie in the range 672–689 nm. The molar extinction values of the dyes in methanol are rather high in the range (0.78–1.59) × $10^5$ M$^{-1}$ cm$^{-1}$. The spectra of the dyes at the studied concentration (1 µM) in the mentioned organic solvents have close shapes and are expected to belong to the monomeric form of the dyes.

At the same time, in the absorption spectra of **dyes 1–4** in 50 mM Tris–HCl buffer (pH 7.9), the strong short-wavelength shift of the main maximum to 590–601 nm was observed (figure 2*a*, table 1). Such hypsochromic shift of the maximum is a hallmark of H-aggregates (most possibly dimers), i.e. the 'card pack' structures formed by cyanine dye molecules; such aggregates are usually non- or weakly fluorescent [31–33]. Besides this main maximum attributed to H-aggregates, the band at 655 nm (figure 2*a*, table 1) was also observed in the spectrum of **dye 4**; this band could be connected with the monomers of this dye. As for the other bands manifested as shoulders in the spectra of **dyes 1**, **2** and **3**, they could belong to aggregates containing higher number of monomer molecules (at 525–565 nm for **dyes 1** and **2**), and to either minor long-wavelength bands of H-aggregates, or conformational isomers of dyes (at 715–830 nm for **dyes 1–3**).

To investigate the effect of pH on the dyes properties, absorption spectra of the dyes were also acquired in 0.05 M phosphate buffer (pH 6.0) and in 0.05 M Tris–HCl (pH 9.0) (electronic supplementary material, figure S2). It was shown that the absorption spectra of the dyes at pH 6.0

have the same shape as the spectra of corresponding dyes at pH 7.9. However, the transition to pH 9.0 results in the changes of the shape of absorption spectra for **dyes 1** and **2** (and to some extent **dye 4**). These changes which consist in the redistribution of the relative intensity of short-wavelength bands could be connected with the change in the relative population of different kinds of dye H-aggregates.

The presence of HSA in the dyes solutions leads to a decrease of H-aggregate peaks and the appearance of absorption peaks (672–686 nm) which are close to those observed in organic solvents and could thus be attributed to dye monomers; these peaks dominate in the spectra of all the studied dyes except **dye 3** (figure 2b). This result indicates the destruction of H-aggregates of the dye in the presence of the protein and the binding of dye monomers to HSA.

## 3.2. Spectral-luminescent characteristics of pentamethine cyanine dyes in the presence of different globular proteins

### 3.2.1. Spectral-luminescent properties of free pentamethine cyanine dyes in aqueous buffer

Spectral-luminescent properties of the series of pentamethine cyanine dyes in buffer solution are presented in table 2. The studied dyes possess low to moderate fluorescence intensity in the aqueous 0.05 M Tris–HCl buffer (pH 7.9). Fluorescence maxima of these dyes are located in the far-red area of the spectrum (675–690 nm), while fluorescence excitation maxima are in the range 661–680 nm that is close to the absorption maxima of these dyes in organic solvents. Hence, emission of the dyes in the buffer could be attributed to their monomer molecules which, though outnumbered by non-fluorescent H-aggregates, are the only fluorescent form of the dye in this medium. Thus, the dyes have small shifts between excitation and emission maxima (11–14 nm). The values of fluorescence quantum yield for **dyes 2** and **3** in a free state in 0.05 M Tris–HCl buffer (pH 7.9) are insignificant (0.15 and 0.2%, respectively; table 3).

### 3.2.2. Spectral-luminescent properties of pentamethine cyanine dyes in the presence of serum albumins

The spectral-luminescent properties of the pentamethine cyanines in the presence of BSA, HSA and ESAs are presented in table 2. The addition of SAs leads to the shift of excitation and emission maxima of the dyes to the long-wavelength spectral region for 13–23 nm that points to the binding of the dyes to proteins. Fluorescence excitation and emission maxima of the dyes in the presence of SAs fall thus in the range 681–793 nm and 694–715 nm, respectively, being in the far-red region of the spectrum. This binding also results in a significant increase in fluorescence intensity of the studied dyes. The most significant spectral response on albumins was observed for **dye 3** (based on benzothiazole and naphtothiazole heterocycles and bearing sulfonate group). The fluorescence intensity of **dye 3** increases in the presence of BSA, HSA and ESA by 79, 161 and 126 times, respectively (table 2, figure 3a). Other studied dyes are also found to be sensitive to SAs; they demonstrate an increase of emission intensity 16–49 times in the presence of these proteins. It should be mentioned that the highest (compared to other dyes) values of the intensity enhancement for **dye 3** are owing to the lowest intensity of this dye in the free state. It could be also noted that while **dyes 1**, **2** and **3** demonstrate the highest intensity in the presence of HSA as compared to other SAs, **dye 4** gives the highest fluorescent response on the ESA protein. As could be guessed from the strong increase in dyes' fluorescence intensity, the binding of the dyes to SAs also leads to strong enhancement in fluorescence quantum yield of these dyes. Thus, the quantum yield value reaches 42% for **dye 3** in 0.05 M Tris–HCl buffer (pH 7.9) in the presence of HSA (increase of 210 times compared to the free dye in the same buffer), and 15% for methylenedioxibenzothiazole **dye 2** in 0.05 M Tris–HCl buffer (pH 7.9) in the presence of BSA (100 times) (table 3).

The dynamic range and lower limit of HSA detection by pentamethine cyanine **dye 3** was determined *via* the titration of a fixed concentration of the dye (5 µM) with increasing amounts of HSA. The lower limit of detection could be determined as 0.004 mg ml$^{-1}$ of HSA, the presence of which leads to about an 8 times increase in the dyes fluorescence intensity over that of the free dye. Further addition of protein resulted in monotonous enhancement of **dye 3** emission intensity up to the protein concentration of 2 mg ml$^{-1}$. Hence the range of HSA detection by pentamethine cyanine **dye 3** (for 5 µM dye concentration) could be considered to be from 0.004 to 2 mg ml$^{-1}$ of this albumin, indicating the high sensitivity of this dye to HSA (figure 3b).

**Table 2.** Spectral-luminescent properties of cyanine dyes in free state and in the presence of BSA, HSA and ESA. ($\lambda_{ex}$ ($\lambda_{em}$), maximum wavelength of fluorescence excitation (emission) spectrum; $I$, emission intensity of dye in free state and in the presence of bovine serum albumin (BSA), human serum albumin (HSA) and horse (equine) serum albumin (ESA) (in arb. units, arbitrary units).)

| dye | free state | | | with BSA | | | with HSA | | | with ESA | | |
| --- | --- | --- | --- | --- | --- | --- | --- | --- | --- | --- | --- | --- |
| | $\lambda_{ex}$, nm | $\lambda_{em}$, nm | $I$, arb. units | $\lambda_{ex}$, nm | $\lambda_{em}$, nm | $I$, arb. units | $\lambda_{ex}$, nm | $\lambda_{em}$, nm | $I$, arb. units | $\lambda_{ex}$, nm | $\lambda_{em}$, nm | $I$, arb. units |
| ANS | 374 | 464 | 4 | 376 | 470 | 418 | 376 | 472 | 328 | 375 | 470 | 168 |
| **dye 1** | 664 | 675 | 103 | 687 | 695 | 1668 | 684 | 694 | 2886 | 686 | 697 | 1941 |
| **dye 2** | 680 | 694 | 48 | 702 | 711 | 1296 | 694 | 707 | 2227 | 703 | 715 | 1000 |
| **dye 3** | 669 | 680 | 11 | 689 | 698 | 870 | 689 | 700 | 1773 | 687 | 699 | 1382 |
| **dye 4** | 661 | 675 | 24 | 681 | 694 | 888 | 682 | 695 | 680 | 683 | 694 | 1182 |

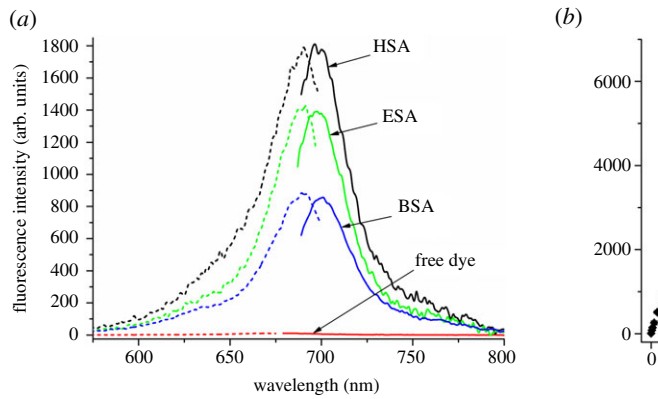

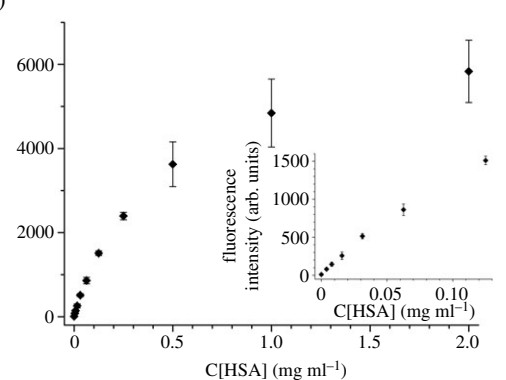

**Figure 3.** (*a*) Fluorescence emission (right) and excitation (left) spectra of **dye 3** in unbound state and in the presence of serum albumins BSA, HSA and ESA. (*b*) The dependence of fluorescence intensity (in arbitrary units—arb. units) of the pentamethine cyanine **dye 3** (5 µM) on HSA concentration, demonstrating the range of HSA detection by this dye. Experiment performed three times, standard deviation presented as error bars.

**Table 3.** Quantum yield values of most efficient dyes in 0.05 M Tris–HCl buffer, pH 7.9.

| dye | quantum yield | | |
| --- | --- | --- | --- |
| | free state (%) | in HSA presence (%) | in BSA presence (%) |
| **dye 2** | 0.15 | — | 15 |
| **dye 3** | 0.2 | 42 | — |

### 3.2.3. Spectral-luminescent properties of pentamethine cyanine dyes in the presence of ovalbumin and β-lactoglobulin

Because the studied far-red dyes were found to be sensitive to SAs, it is important to also study their spectral-fluorescent response to the proteins of other classes; this could clarify the question of the dyes selectivity to SAs. Thus, the spectral properties of the studied dyes in the presence of globular proteins with different structures such as OVA and BLG were analysed (higher concentrations of OVA and BLG were used as compared to those of SAs). OVA is a member of the serpin superfamily without inhibitory activity. Its native structure is mainly comprised a mixture of α-helixes and β-sheets, and a flexible loop–helix–loop motif constitutes the reactive centre [34]. BLG is a lipocalin protein that folds up into an 8-stranded, antiparallel β-barrel with a 3-turn α-helix on the outer surface and a ninth β-strand flanking the first strand [35]. The lipocalin family includes, among others, transport proteins such as the retinol-binding protein, the odorant-binding protein and the major urinary protein, which all share the common structural feature of a β-barrel calyx built from eight antiparallel *β* sheets, arranged as an ideal site for hydrophobic ligands [36].

Spectral-fluorescent characteristics of the studied dyes in the presence of OVA and BLG are presented in table 4. As in the case of SAs, the presence of OVA and BLG causes the long-wavelength shift of the maxima of the dyes' fluorescence excitation and emission spectra for 14–22 nm (except for **dyes 1** and **3** in the presence of BLG where these shifts are negligible). At the same time, all the studied dyes demonstrate notably lower fluorescence intensity in the presence of BLG and OVA as compared to SAs (tables 3 and 4). Thus, the addition of OVA and BLG results in an increase of the dyes fluorescence intensity by 5–24 times, while in the presence of SAs this increase amounted to 16–161 times. The most prominent difference in fluorescent response to BLG and OVA (intensity increase by 23 and 24 times, respectively) versus SAs (79–161 times) was observed for **dye 3**; this dye was shown above to increase its fluorescence quantum yield value up to 200 times in the presence of HSA (table 3). Hence, this dye has weak responses on proteins with structures different from those of SAs. This makes **dye 3** promising for future use as a fluorescent probe for SAs.

We have also compared the fluorescent response of the studied far-red dyes with that of the widely used fluorescent hydrophobic protein probe 1,8-ANS (figure 1). For this dye, biding to protein is known

**Table 4.** Spectral-luminescent properties of cyanine dyes in free state and in the presence of beta-lactoglobulin and ovalbumin. ($\lambda_{ex}$ ($\lambda_{em}$), maximum wavelength of fluorescence excitation (emission) spectrum; $I$, emission intensity of dye in free state and in the presence of OVA and BLG (in arb. units—arbitrary units).)

| dye | free dye | | | OVA | | | BLG | | |
|---|---|---|---|---|---|---|---|---|---|
| | $\lambda_{ex}$, nm | $\lambda_{em}$, nm | $I$, arb. units | $\lambda_{ex}$, nm | $\lambda_{em}$, nm | $I$, arb. units | $\lambda_{ex}$, nm | $\lambda_{em}$, nm | $I$, arb. units |
| ANS | 374 | 464 | 4 | 375 | 465 | 8 | 376 | 470 | 12 |
| **dye 1** | 664 | 675 | 103 | 684 | 694 | 508 | 665 | 676 | 687 |
| **dye 2** | 680 | 694 | 48 | 702 | 712 | 212 | 701 | 709 | 615 |
| **dye 3** | 669 | 680 | 11 | 688 | 697 | 258 | 671 | 678 | 248 |
| **dye 4** | 661 | 675 | 24 | 676 | 689 | 311 | 680 | 692 | 475 |

to occur *via* the combination of hydrophobic interaction i.e. burying of the dye into the hydrophobic site of the protein [37] and electrostatic interaction between the negatively charged sulfonate group of ANS with positively charged amino acids (e.g. histidine, lysine or arginine) [11].

Spectral-fluorescent characteristics of 1,8-ANS solution in buffer in free form and in the presence of the studied proteins (with the concentrations of dye and proteins equal to those used for the studied far-red dyes) are provided in tables 2 and 4. Insignificant long-wavelength shifts of fluorescence excitation and emission maxima (up to 8 nm) are observed for ANS in the presence of all proteins. At the same time, while emission intensity of free ANS is insignificant (4 arb. units), this dye increases its fluorescence intensity in the presence of SAs by 42–105 times (table 2) as compared to 2 and 3 times in the presence of OVA and BLG, respectively (table 4). However, the values of ANS emission intensity in complexes with SAs is several times lower as compared to the studied pentamethine cyanines (168–418 arb. units for ANS versus 680–2886 arb. units for the studied far-red dyes); hence ANS is a 'less bright' probe when bound to SAs as compared to the studied far-red dyes.

It was shown that a single BSA globule (at pH values between 5 and 10) can bind five molecules of 1,8-ANS resulting in their fluorescent response [10,11]. At the same time, according to literature data, the BLG molecule has two binding sites for ANS. One of them is located on a surface hydrophobic patch, while another binding site is located inside the protein calyx [38]. Comparing the intensity of the fluorescent response of ANS on the presence of studied proteins to this of pentamethine dyes, we could suppose that higher fluorescent sensitivity of pentamethines to SAs as compared to other proteins could be partially explained (together with possibly higher values of the binding constants) by the higher number of appropriate binding sites on SA globules over that on BLG or OVA ones.

## 3.3. Molecular docking

For the dyes with a wide chromophore system, the non-specific hydrophobic interaction is suggested to be a preferred mode of binding with proteins [18]. Because the reported far-red dyes possess higher fluorescent sensitivity to SAs as compared to other studied proteins, we suggested that there are also specific interactions of these dyes with the binding sites of SAs. To estimate the possible binding places on SAs, we have performed molecular docking of four studied pentamethine cyanine dyes to the entire protein surface of HSA. Because of the high sensitivity of **dye 4** to ESA, molecular docking of **dye 4** for this albumin was also performed.

SAs consist of about 580 amino acid residues, nearly 67% of which are contained in α-helices; the globular structure of SAs also includes six turns along with 17 disulfide bridges in a reiterating sequence of nine loops. The tertiary structure of SAs consists of three domains I, II and III of the analogous structure, each containing A and B subdomains (composed of 4 and 6 α-helices, respectively). HSA and BSA show approximately 80% of structural homology and a repetitive structure of disulfide bonds [39]. SAs are considered to contain several binding sites for various molecules. Thus, in the tertiary structure of HSA besides two principal drug-binding sites I and II (in subdomains IIA and IIIA, respectively), several other sites were found [40]. Particularly, seven fatty acid-binding sites were reported in [41]; one of them (in subdomain IB) could be also occupied by haemin [42].

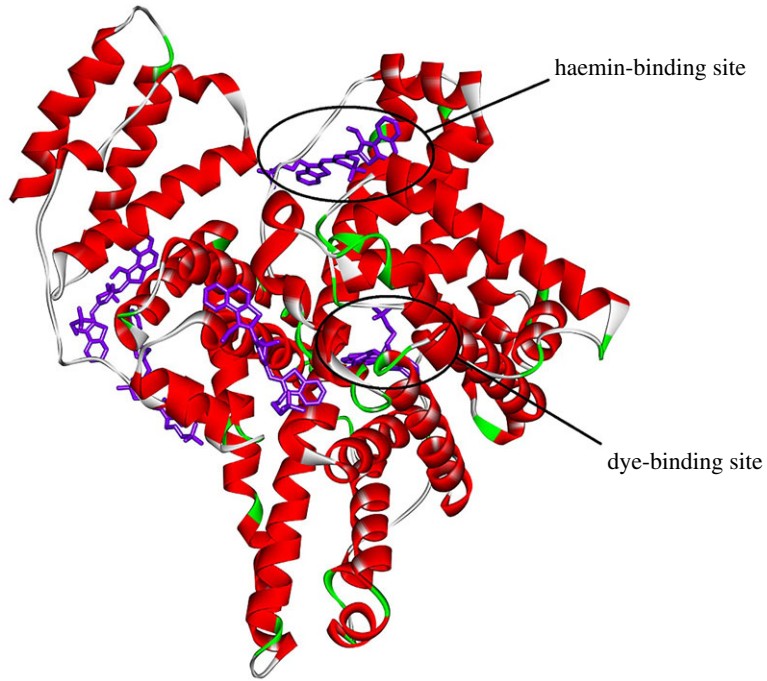

**Figure 4.** The complex of HSA with fluorescent **dye 3**, obtained with molecular docking to the entire protein surface.

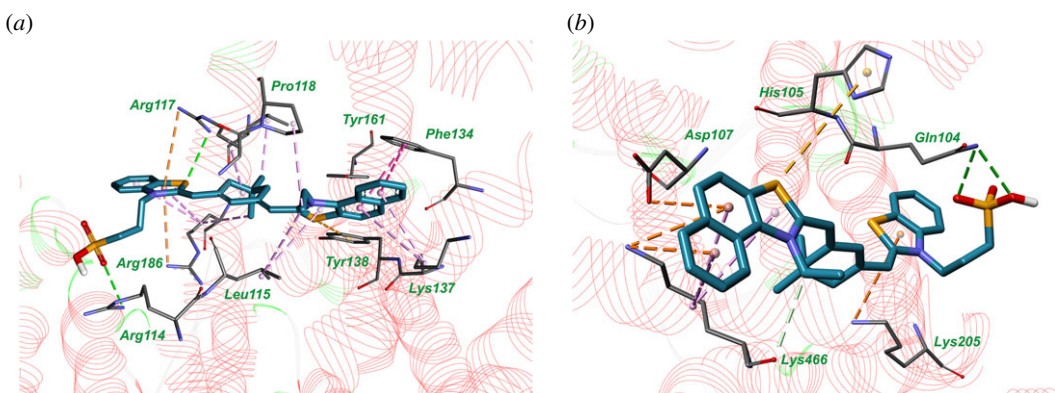

**Figure 5.** The complex of fluorescent **dye 3** with haemin-binding site (*a*) and dye-binding site (*b*), obtained with molecular docking. Hydrogen bonds are indicated with green dotted lines and hydrophobic interactions are represented by magenta dotted lines.

Molecular docking results have shown that HSA has five binding sites which are common for all investigated dyes (figure 4). The best score (pointing to the most energetically favourable electrostatic and Van der Waals interactions) was obtained for the haemin-binding site and for the dye-binding site in the pocket between subdomains IA, IIA and IIIA (this site does not match for any of seven fatty acid-binding sites). The complex of the compound **dye 3** with amino acid residues of the haemin-binding site and dye-binding site is represented in figure 5. It should be mentioned that if HSA globule has several binding sites for **dye 3** (as it is shown in figure 5), this could make the results of the fluorescent detection of albumin by this dye more sensitive to additional factors such as temperature or presence of other compounds; this should be taken into account in the case of practical application of **dye 3**.

As mentioned above, the fluorescent **dye 4** has a higher affinity to ESA than to HSA. According to the results of molecular docking to the whole surface of human and ESAs, this fluorescent probe has different preferential binding sites in equine and HSAs (figure 6*a,b*). We have performed superposition of HSA and ESAs (figure 6*c*) and revealed that three-dimensional structures of these proteins are quite similar but have a significant difference in the size of their haemin-binding sites (around the dye). Thus, the possible reason of **dye 4** binding to different pockets in ESA and HSA could be the smaller size of the haemin-binding site in the former protein as compared to the latter one.

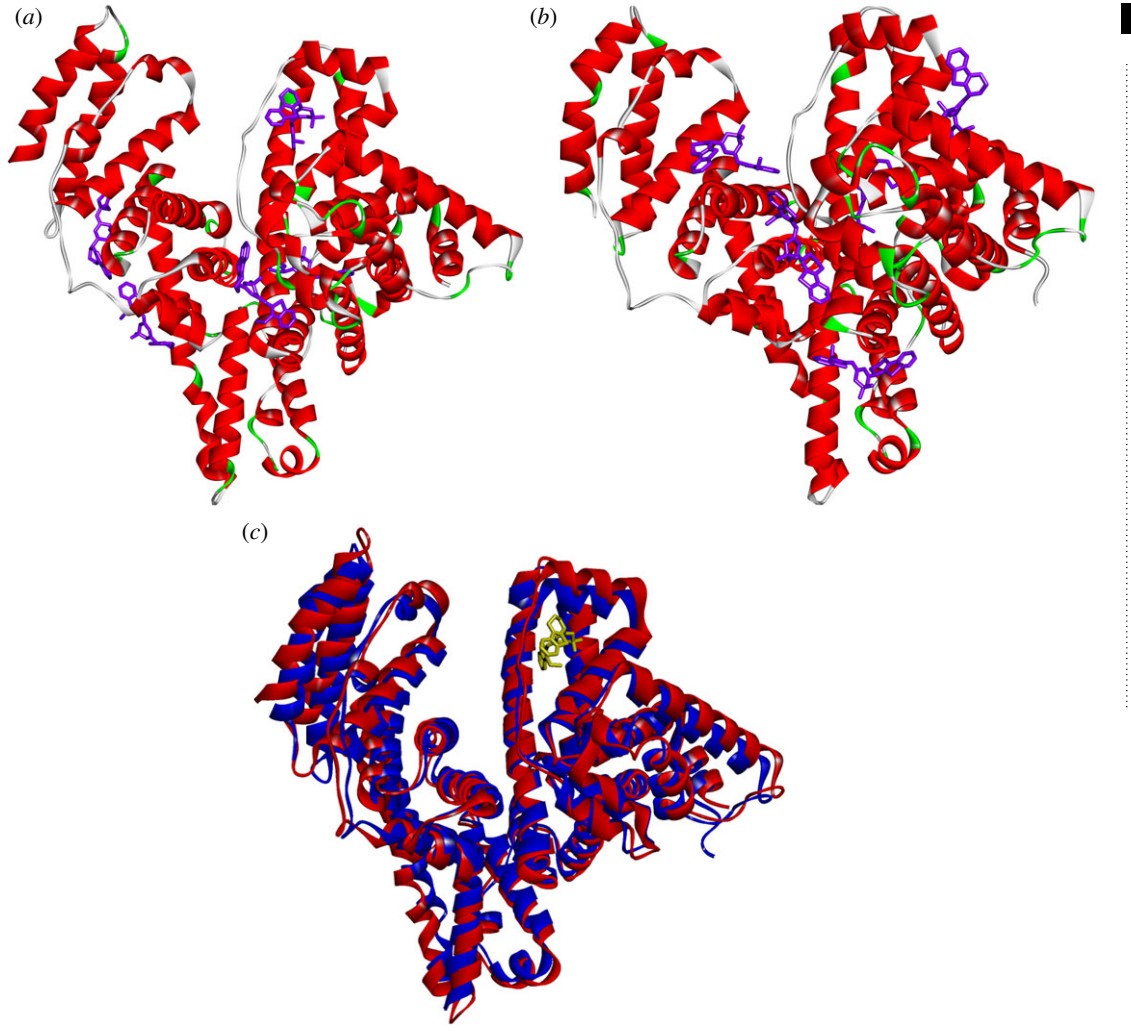

**Figure 6.** The complexes of fluorescent **dye 4** with the whole surface of human (*a*) and equine (*b*) serum albumin, obtained with molecular docking. (*c*) The superposition of human (red colour) and equine (blue colour) serum albumins with the dye (yellow colour) in the haemin-binding site of HSA.

## 3.4. Estimation of the affinity of **dye 3** binding to human serum albumin

Basing on the obtained dependence of fluorescence intensity of **dye 3** on HSA concentration, we have estimated the equilibrium constant of **dye 3** binding with HSA (figure 7). For this, the experimental plot was fitted by equation (2.5), obtained with the assumption that each protein globule has $n$ sites for dye binding with the equal values of the binding constant $K$. As the result of the fitting with such an equation, it turned out that no reasonable values of $n$ could be obtained as an approximation parameter. This could mean that the used model of $n$ binding sites with equal binding constants per globule does not adequately describe the binding. At the same time, to characterize the binding affinity, the binding constant upon the fixed value on $n = 1$ was estimated as $K = (2.9 \pm 0.4) \times 10^5 \, \mathrm{M}^{-1}$. Meanwhile, if based on molecular docking data we consider each globule to possess five binding sites (fixed value of $n = 5$), the fitting gives the binding constant value of $K = (3.0 \pm 0.1) \times 10^4 \, \mathrm{M}^{-1}$. It is seen from figure 7 that, though the value of $n$ cannot be obtained as an approximation parameter with the used method, the fitting is more adequate in the case on fixed value $n = 5$ as compared to $n = 1$. We could thus suppose, that the HSA globule has several sites for the binding of **dye 3**.

## 4. Conclusion

We observed that all far-red dyes studied in this work exhibit a stronger fluorescent 'light-up' response in the presence of SAs (BSA, HSA and ESA) than in the presence of other globular proteins (OVA and BLG).

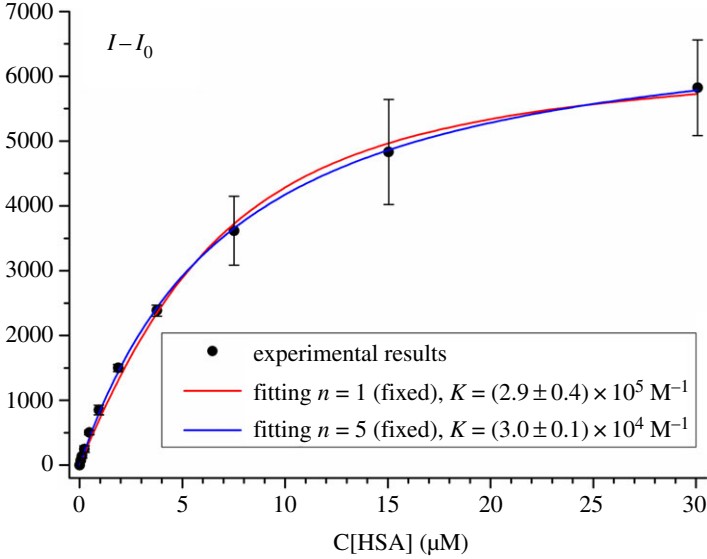

**Figure 7.** Dependence of $I - I_0$ on HSA concentration ($I$ and $I_0$ are fluorescence intensities of **dye 3** in the presence of certain HSA concentration and of the free dye, respectively, the standard deviation is presented as error bars), and fitting of this dependence by equation (2.5) with values of $n$ fixed as $n = 1$ and $n = 5$. Dye concentration 5 µM.

The most efficient **dye 3** demonstrates the 79-, 161- and 126-fold increase of the fluorescent intensity in the presence of BSA, HSA and ESA, respectively. The values of fluorescence quantum yield were found to be equal to 15% for **dye 2** complexed with BSA and 42% for **dye 3** bound to HSA, while quantum yields for the free **dyes 2** and **3** were 0.15% and 0.2%, respectively. The range of HSA detection by **dye 3** was estimated as 0.004–2 mg ml$^{-1}$, which indicates the high sensitivity of the dye to this protein. Estimation of the affinity of **dye 3** binding to HSA has shown that the protein globule probably has several binding sites for this dye. According to the results of molecular docking, all studied dyes bind to up to five binding pockets on HSA, the most preferable of them are identified as the haemin-binding site and the site located in the pocket between subdomains IA, IIA and IIIA. Thus, the studied pentamethine cyanine dyes are suggested as promising high-efficient far-red probes for SAs detection owing to their intensive spectral response and notable fluorescent sensitivity to these proteins.

Data accessibility. Data are available from the Dryad Digital Repository: https://doi.org/10.5061/dryad.s7h44j13s [43].
Authors' contributions. D.A., S.C. and M.L. carried out laboratory experiments, performed the data analysis and drafted the manuscript; G.V. performed molecular docking, co-wrote the manuscript; A.B. and Yu.S. designed and synthesized cyanine dyes, performed NMR, co-wrote the manuscript; A.M., S.Y. and V.K. designed the study and coordinated it, and critically revised the manuscript. All authors gave final approval for publication and agreed to be held accountable for the work performed therein.
Competing interests. We declare we have no competing interests.
Acknowledgements. The project NoBiasFluors leading to these results has received funding from the European Union's Horizon 2020 research and innovation programme under the Marie Skłodowska-Curie grant agreement no. 872331. S.C. and G.V. are grateful to the grant of a group of young scientists no. 0120U000079 for 2020–2021.

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
