## [Reviewer comments · Royal Society Open Science]

Review History

RSOS-200453.R0 (Original submission)

Review form: Reviewer 1

Is the manuscript scientifically sound in its present form?

No

Are the interpretations and conclusions justified by the results?

No

Is the language acceptable?

Yes

Do you have any ethical concerns with this paper?

No

Have you any concerns about statistical analyses in this paper?

No

Recommendation?

Major revision is needed (please make suggestions in comments)

Comments to the Author(s)

In this work, the authors observed that fluorescence due to bridged thiadicarbocyanine dyes intensified with the amount of co-existing serum albumins. These dicarbocyanine dyes are allegedly promising high-efficient far-red probes for serum albumin detection, especially useful for biological samples.

Bridged dicarbocyanine dyes are well known compounds. It is also well known that fluorescence of cyanine dyes intensifies in the presence of serum albumins. This paper therefore presents just common properties of common compounds. The authors should elaborate on the advances made by using bridged dicarbocyanine dyes in comparison with other far-red to near infra red cyanine dyes.

My concerns the authors should address are listed below.

1. The authors should demonstrate the advances made by using the bridged dicarbocyanine dyes in comparison with other far-red to near infra-red cyanine dyes.

Enhancement of fluorescence of cyanine dye by serum albumin has been reported, for example: G. Patonay et al. "Spectroscopic Study of a Bis(heptamethine cyanine)dye and Its Interaction with Human Serum Albumin" *Appl. Spectrosc.* 59, 682-90 (2005); M. Saikiran et al. "Photophysical investigation of squaraine and cyanine dyes and their interaction with bovine serum albumin", *J. Physics: Conference Series*, 2016, 704, 012012, doi:10.1088/1742-6596/704/1/012012.

Bridged dicarbocyanine dyes are quite common especially in the field of silver halide photographic materials: US5576173, for example.

2. Structures and fluorescence spectra of albumin-bound dyes should be clarified.

As shown in Fig. 5, albumin-bound cyanine dyes in different binding site have different conformation. They must show different fluorescence spectra. The dye molecule in Fig. 5b distorted considerably from a planar all-trans conformation, for example. Equation (3), which assumes that all the bound dyes have the same fluorescence intensity, cannot therefore apply to the present case. Albumin complex with dyes of various fluorescence intensities would make the calibration curve dependent on such factors as temperature, coexisting compounds etc. This dependence would impair the reliability of the quantitative analysis.

3. Purity of Dye 1 should be checked.

Molar extinction coefficient of Dye 1 is exceptionally low even in the methanol solution, in which Dye 1 is probably monomeric. (Table 1).

4. Fig. 1: The chemical formula of 1,8-ANS is incomplete. Probably one proton is missing.

Review form: Reviewer 2

Is the manuscript scientifically sound in its present form?

Yes

Are the interpretations and conclusions justified by the results?

Yes

Is the language acceptable?

Yes

Do you have any ethical concerns with this paper?

No

Have you any concerns about statistical analyses in this paper?

Yes

Recommendation?

Accept with minor revision (please list in comments)

Comments to the Author(s)

The authors developed a far-red pentamethine cyanine dyes as fluorescent probes for detection of serum albumins. It is interesting. Publication is recommended after minor revisions. 1. The authors have briefly introduced the advantage of fluorescent detection in the introduction. Some recent representative progress should be cited here: *Angew. Chem. Int. Ed.*, 2016, 55, 12751-12754; *Angew. Chem. Int. Ed.* 2017, 56, 16611-16615; *J Am Chem Soc.* 2019,141(5):2061-2068;*Angew. Chem. Int. Ed.* 2019, 58, 16067-16070. 2. The authors listed some quantum yields (QY) of dyes, the corresponding medium of the QY should be listed in main text. 3. The authors should test the effect of pH and different solvents for the dyes.4 The error bars should be completed in the figure.

Decision letter (RSOS-200453.R0)

15-Apr-2020

Dear Dr Kovalska:

Title: Far-red pentamethine cyanine dyes as fluorescent probes for detection of serum albumins
Manuscript ID: RSOS-200453

The editor assigned to your manuscript has now received comments from reviewers. We would like you to revise your paper in accordance with the referee and Subject Editor suggestions which can be found below (not including confidential reports to the Editor). Please note this decision does not guarantee eventual acceptance.

Please submit your revised paper before 08-May-2020. Please note that the revision deadline will expire at 00.00am on this date. If we do not hear from you within this time then it will be assumed that the paper has been withdrawn. In exceptional circumstances, extensions may be possible if agreed with the Editorial Office in advance. We do not allow multiple rounds of revision so we urge you to make every effort to fully address all of the comments at this stage. If deemed necessary by the Editors, your manuscript will be sent back to one or more of the original reviewers for assessment. If the original reviewers are not available we may invite new reviewers.

RSC Associate Editor:
Comments to the Author:
(There are no comments.)

RSC Subject Editor:
Comments to the Author:
(There are no comments.)

Reviewers' Comments to Author:
Reviewer: 1

Comments to the Author(s)

In this work, the authors observed that fluorescence due to bridged thiadicarbocyanine dyes intensified with the amount of co-existing serum albumins. These dicarbocyanine dyes are allegedly promising high-efficient far-red probes for serum albumin detection, especially useful for biological samples.

Bridged dicarbocyanine dyes are well known compounds. It is also well known that fluorescence of cyanine dyes intensifies in the presence of serum albumins. This paper therefore presents just common properties of common compounds. The authors should elaborate on the advances made by using bridged dicarbocyanine dyes in comparison with other far-red to near infra red cyanine dyes.

My concerns the authors should address are listed below.

1. The authors should demonstrate the advances made by using the bridged dicarbocyanine dyes in comparison with other far-red to near infra-red cyanine dyes.

Enhancement of fluorescence of cyanine dye by serum albumin has been reported, for example: G. Patonay et al. "Spectroscopic Study of a Bis(heptamethine cyanine)dye and Its Interaction with Human Serum Albumin" *Appl. Spectrosc.* 59, 682-90 (2005); M. Saikiran et al. "Photophysical investigation of squaraine and cyanine dyes and their interaction with bovine serum albumin", *J. Physics: Conference Series*, 2016, 704, 012012, doi:10.1088/1742-6596/704/1/012012.

Bridged dicarbocyanine dyes are quite common especially in the field of silver halide photographic materials: US5576173, for example.

2. Structures and fluorescence spectra of albumin-bound dyes should be clarified.

As shown in Fig. 5, albumin-bound cyanine dyes in different binding site have different conformation. They must show different fluorescence spectra. The dye molecule in Fig. 5b distorted considerably from a planar all-trans conformation, for example. Equation (3), which assumes that all the bound dyes have the same fluorescence intensity, cannot therefore apply to the present case. Albumin complex with dyes of various fluorescence intensities would make the calibration curve dependent on such factors as temperature, coexisting compounds etc. This dependence would impair the reliability of the quantitative analysis.

3. Purity of Dye 1 should be checked.

Molar extinction coefficient of Dye 1 is exceptionally low even in the methanol solution, in which Dye 1 is probably monomeric. (Table 1).

4. Fig. 1: The chemical formula of 1,8-ANS is incomplete. Probably one proton is missing.

Reviewer: 2

Comments to the Author(s)

The authors developed a far-red pentamethine cyanine dyes as fluorescent probes for detection of serum albumins. It is interesting. Publication is recommended after minor revisions. 1. The authors have briefly introduced the advantage of fluorescent detection in the introduction. Some recent representative progress should be cited here: *Angew. Chem. Int. Ed.*, 2016, 55, 12751-12754; *Angew. Chem. Int. Ed.* 2017, 56, 16611-16615; *J Am Chem Soc.* 2019,141(5):2061-2068; *Angew. Chem. Int. Ed.* 2019, 58, 16067-16070. 2. The authors listed some quantum yields (QY) of dyes, the corresponding medium of the QY should be listed in main text. 3. The authors should test the effect of pH and different solvents for the dyes. 4 The error bars should be completed in the figure.

Author's Response to Decision Letter for (RSOS-200453.R0)

See Appendix A.

Decision letter (RSOS-200453.R1)

Dear Dr Kovalska:

Title: Far-red pentamethine cyanine dyes as fluorescent probes for detection of serum albumins
Manuscript ID: RSOS-200453.R1

It is a pleasure to accept your manuscript in its current form for publication in Royal Society Open Science. The chemistry content of Royal Society Open Science is published in collaboration with the Royal Society of Chemistry.

RSC Associate Editor
Comments to the Author:
(There are no comments.)

Reviewer(s)' Comments to Author:

Appendix A

Dear Dr. Laura Smith,

Thank you for your kind attention to our manuscript, and I am also thankful to the referees for their valuable remarks. We did our best to account for all of the reviewers' comments. Our answers to the comments of the referees are as follows:

Reviewer: 1

1. The authors should demonstrate the advances made by using the bridged dicarbocyanine dyes in comparison with other far-red to near infra-red cyanine dyes.

Enhancement of fluorescence of cyanine dye by serum albumin has been reported, for example: G. Patonay et al. "Spectroscopic Study of a Bis(heptamethine cyanine)dye and Its Interaction with Human Serum Albumin" Appl. Spectrosc. 59, 682-90 (2005); M. Saikiran et al. "Photophysical investigation of squaraine and cyanine dyes and their interaction with bovine serum albumin", J. Physics: Conference Series, 2016, 704, 012012, doi:10.1088/1742-6596/704/1/012012.

Bridged dicarbocyanine dyes are quite common especially in the field of silver halide photographic materials: US5576173, for example.

- We have extended the Introduction, and added some discussion about the used NIR cyanine dyes and pentamethine cyanine dyes (including bridged ones); the articles referred by the Reviewer were discussed. Particularly, based on the literature, the advantages of bridged pentamethine cyanines such as thermostability and higher fluorescence quantum yield were mentioned.

2. Structures and fluorescence spectra of albumin-bound dyes should be clarified.

As shown in Fig. 5, albumin-bound cyanine dyes in different binding site have different conformation. They must show different fluorescence spectra. The dye molecule in Fig. 5b distorted considerably from a planar all-trans conformation, for example. Equation (3), which assumes that all the bound dyes have the same fluorescence intensity, cannot therefore apply to the present case. Albumin complex with dyes of various fluorescence intensities would make the calibration curve dependent on such factors as temperature, coexisting compounds etc. This dependence would impair the reliability of the quantitative analysis.

- We agree with the reviewer in that if a protein globule possesses several (more than one) site for the dye binding, the dyes bound to different sites could give different characteristics of fluorescence spectrum (i.e. maximum wavelength and quantum yield); this could be due to different conformation and degree of internal motions restriction of the dyes bound in different sites. Meanwhile, our measurements showed that the shape and position of maximum of the dye fluorescence spectrum does not depend on HSA concentration; thus, either the protein-bound dye molecules have similar conformation, or the molecules with other conformations make negligible contribution into the fluorescence spectrum. It is still possible that dye molecules bound to different sites have the same conformation but different quantum yield due to different fixation strength. But (having in mind the mentioned complications) for the rough estimation of the binding constant we consider the dye fluorescence intensity to be proportional to the number of the bound dye molecules. The above considerations were added to the Materials and Methods part of the manuscript.

As for the impairing of the reliability of the detection due to the presence of several binding sites, we have pointed this in the Molecular Docking subsection of the Results and Discussion section. At the same time, it should be mentioned that (1) this problem exists for many other probes as well, e.g. ANS has several binding sites, and (2) results of molecular docking account only for electrostatic and Van der Waals' interactions.

3. Purity of Dye 1 should be checked. Molar extinction coefficient of Dye 1 is exceptionally low even in the methanol solution, in which Dye 1 is probably monomeric. (Table 1).

- Molar extinction coefficients were carefully rechecked; rather wide distribution of extinction values was found; corrected values were added into the article. It is worth mentioning that different extinction coefficients can be explained by different structures of dyes which chromophores containing Benzothiazole, Naphthothiazole, dihydrothia-aza-fluorene or [1,3]Dioxolobenzothiazole fragments.

4. Fig. 1: The chemical formula of 1,8-ANS is incomplete. Probably one proton is missing.

- The formula of ANS was corrected

Reviewer: 2

1. The authors have briefly introduced the advantage of fluorescent detection in the introduction. Some recent representative progress should be cited here: Angew. Chem. Int. Ed., 2016, 55, 12751-12754; Angew. Chem. Int. Ed. 2017, 56, 16611-16615; J Am Chem Soc. 2019, 141(5):2061-2068; Angew. Chem. Int. Ed. 2019, 58, 16067-16070.

- We have added an information about these studies to the Introduction.

2. The authors listed some quantum yields (QY) of dyes, the corresponding medium of the QY should be listed in main text.

- Both in Materials and Methods as well as in Results and discussion parts, it was indicated that the QY of the dyes (both free and in the presence of HSA and BSA) was measured in 0.05 M Tris-HCl buffer (pH 7.9) as medium.

3. The authors should test the effect of pH and different solvents for the dyes.

- In addition to the 0.05 M Tris-HCl buffer (pH 7.9), we have additionally studied the dyes acquiring absorption spectra in the 0.05 M phosphate buffer (pH 6.0) and 0.05 M Tris-HCl buffer (pH 9.0). The absorption spectra were added to Supplementary information and discussed in the text. We have also acquired absorption spectra in different solvents (acetonitrile and DMSO, in addition to methanol), new data was added to the Table 1, and spectra provided in Supplementary information.

4 The error bars should be completed in the figure.

- We have repeated the experiment three times, calculated average values with standard deviation, and modified Figs. 3b and 7 with the obtained average values together with standard deviation as error bars; corresponding corrections were also made in text.

With my best regards,

Sincerely

Vladyslava B. Kovalska, Dr.Sci.,
Institute of Molecular Biology and Genetics of NAS of Ukraine.